# Semantic Description of Explainable Machine Learning Workflows for Improving Trust

**Patricia Inoue Nakagawa *** [ID]**, Luís Ferreira Pires** [ID]**, João Luiz Rebelo Moreira** [ID]**,
Luiz Olavo Bonino da Silva Santos** [ID] **and Faiza Bukhsh**

Faculty of Electrical Engineering, Mathematics and Computer Science (EEMCS), University of Twente,
7522 NB Enschede, The Netherlands; l.ferreirapires@utwente.nl (L.F.P.); j.luizrebelomoreira@utwente.nl (J.L.R.M.);
l.o.boninodasilvasantos@utwente.nl (L.O.B.d.S.S.); f.a.bukhsh@utwente.nl (F.B.)
* Correspondence: patyinoue@gmail.com

**Abstract:** Explainable Machine Learning comprises methods and techniques that enable users to better understand the machine learning functioning and results. This work proposes an ontology that represents explainable machine learning experiments, allowing data scientists and developers to have a holistic view, a better understanding of the explainable machine learning process, and to build trust. We developed the ontology by reusing an existing domain-specific ontology (ML-SCHEMA) and grounding it in the Unified Foundational Ontology (UFO), aiming at achieving interoperability. The proposed ontology is structured in three modules: (1) the general module, (2) the specific module, and (3) the explanation module. The ontology was evaluated using a case study in the scenario of the COVID-19 pandemic using healthcare data from patients, which are sensitive data. In the case study, we trained a Support Vector Machine to predict mortality of patients infected with COVID-19 and applied existing explanation methods to generate explanations from the trained model. Based on the case study, we populated the ontology and queried it to ensure that it fulfills its intended purpose and to demonstrate its suitability.

**Keywords:** XAI; machine learning; semantic web technologies; ontology

## 1. Introduction

Artificial Intelligence (AI) and particularly Machine Learning (ML) have been extensively explored due to their ability to learn and perform autonomous tasks, and the potential to achieve better results than humans [1,2]. Among ML models, there are inherently intelligible algorithms, as opposed to inscrutable ones. Models are inherently intelligible to the degree that a human can predict how a change to a feature in the input can affect the output [2]. Inscrutable models are more complex and harder to explain, therefore it is more challenging to understand the reason for their results. Examples are complex neural networks or deep learning. For this reason, these models are often considered black-boxes [3].

To cope with this, Explainable Artificial Intelligence (XAI) considers methods and techniques to make the results of AI systems explainable, intelligible, transparent, interpretable, or comprehensible to humans [1]. ML explainability is relevant because it allows the identification of the changes and optimization of the ML model necessary to generate the results, since being able to understand the model allows us to identify problems and improve the model. This ensures that the system acts adequately, improving trust and avoiding unethical issues [3].

Semantic Web Technologies (SWT) were initially introduced to make the internet data machine-readable by encoding semantics with the data. In the scope of XAI, these techniques can potentially be applied to ML models and are expected to enable the development of truly explainable AI-systems because they provide semantically interpretable tools and allow reasoning on knowledge resources that can help explain ML systems [4–6].

They usually are adopted as complementary sources of information that enrich the datasets with semantic knowledge, enabling the exploitation of the relationships between concepts and inferences of new knowledge [1]. There are two main categories of exploration approaches regarding how explanations are generated by considering the part of the machine learning process that is using the semantic resource, namely (1) ante-hoc and (2) post-hoc, as schematically depicted in Figure 1.

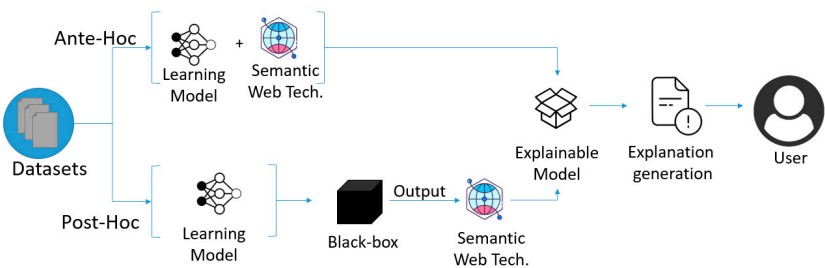

**Figure 1.** ML explanation approaches.

Ante-hoc explainability builds an intrinsic explainable model, using semantic resources during the ML training process to build explainable learning models that generate predictions together with explanations of its reasoning. In this case, the semantic source is integrated intrinsically to the ML algorithm to obtain explanations considering the internal functioning of the model by mirroring the structure of knowledge graphs, using knowledge resources as embeddings, or exploring the ontology taxonomy, among others. The transparency of the ante-hoc model facilitates understanding and enables adequate changes in the ML when necessary [1]. However, changes are necessary to incorporate the background knowledge to the algorithms, forcing design choices and creating a bias towards explainability, which can affect the performance of existing models regarding accuracy and efficiency. Consequently, the solutions are often model-specific and domain-specific, resulting in less generalizable and versatile outcomes [7].

Post-hoc explainability consists of wrapping fully black-box trained models and adding an explainability layer [8]. The biggest advantage in post-hoc solutions is that they are model-agnostic, that is, the explanations are separated from the ML model, thus no change is needed to the ML model so that the solutions can be used across different models. Post-hoc solutions explain the logic of the output by trying to justify the reason why the ML model generates the results. The use of SWT empowers the explainable ML tools by expanding their knowledge without requiring prior experience, creating explanations for patterns or questions that go beyond the data analyzed [1]. Nonetheless, they might not be truthful to the underlying ML algorithm because the explanations result from artifacts that mimic the behavior of the black-box, based on hypotheses that do not take into account the internal functioning of the ML model (e.g., node activations), nor the actual knowledge that the ML model gets from the data, raising concerns related to trust, reliability, and fidelity. Furthermore, most post-hoc solutions focus on local explanations, that is, generating explanations for a single output. Few solutions focus on global explanations, which would clarify the whole performance of the model [7].

Besides considering how to obtain the explanation, in order to evaluate the ML model and verify if it is suitable for its task, we should not only understand its logic but also have an overview of the whole ML process, since it consists of many components that influence the behavior and results of ML models. For example, the data used to train ML algorithms together with the preprocessing steps adopted to enhance the quality of the data have a significant impact on the model's performance, since ML algorithms rely on identifying data patterns or regularities, which may lead these algorithms to follow some possible bias present in the data [9].

Learning is always based on available data, and there may be differences between training data and real data [2]. Small changes in the input can make big differences in

the output, which can lead to serious errors when the system is used in the real world. In addition, the input datasets are often noisy, biased, and sometimes contain incorrectly labeled samples. Without knowing the data quality, training the model is a tricky and challenging task [10]. Therefore, explainability needs to be addressed from the input data step [11].

Furthermore, the evaluation of the ML implementation needs to use adequate measurements according to the task and the application so the user can comprehend the evaluation method to correctly select the most suitable model. For example, in our COVID-19 case study, diagnosis detection models should have high sensitivity, identifying most patients that truly have a condition, and high specificity, avoiding the identification of a condition in patients that do not have it. Hence, an overview of the entire ML process, from the data input to the evaluation, would allow the user to verify if the ML model is adequate by having a better and more complete understanding of the decision process and the reason why the ML model arrived at specific decisions, and identify where to make corrections and adjustments.

The goal of this research is to leverage ML post-hoc explainability by proposing an ontology that represents and provides a holistic overview of the entire ML and post-hoc explanation processes. This enables the user to have a better and more complete understanding of those processes and complements post-hoc explanations that justify the reason why the ML model arrived at specific decisions, enhancing trust.

This paper is structured as follows: Section two provides the ontology concepts, describing all the steps necessary to specify and develop the ontology. Section three presents the case study and the experiments performed to populate the ontology. Section four describes the results obtained, such as the conceptual model and the ontology evaluation. Finally, Section five discusses the results, the contributions, and future work.

## 2. Ontology Concepts

An ontology consists of a collection of related concepts that describe a particular domain, with definitions for objects and types of objects that provide a semantic vocabulary to define the meaning of things. An ontology is made machine-interpretable with knowledge representation techniques so that it can be used by applications to reason about the domain of knowledge [12].

### 2.1. Ontology Specification

The first phase to build an ontology consists of the ontology specification, when the methodology to be adopted as a guideline is chosen, as well as the goal, scope, and requirements to the ontology. This is a preparation stage before ontology development. During the specification phase, knowledge acquisition is also performed to find knowledge sources such as other ontologies, aiming at the reuse of already established conceptualizations and achieving interoperability.

The ontology development of this project follows the guidelines of SABiO (Systematic Approach for Building Ontologies) [13], which proposes a process for the development of domain ontologies based on foundational ontologies. It consists of five main steps and supporting processes that are performed in parallel to the main development process. The five main steps are (1) purpose identification and requirements elicitation; (2) ontology capture and formalization; (3) design; (4) implementation; and (5) test. SABiO also distinguishes reference ontologies from operational ontologies, where reference ontologies are developed in the first two steps and the operational ontologies should follow the design and implementation steps of the process.

### 2.1.1. Ontology Purpose and Requirements

The purpose of developing our domain-specific ontology is to represent the entire ML process and post-hoc explanation process, enabling data scientists to have a holistic view and a better understanding of those processes, aiming to complement and leverage the

post-hoc explanations. The ontology captures the concepts of the domain and makes them machine-interpretable, making it possible to keep track of the steps from the processes and retrieve information from them.

The ontology must comply with functional and non-functional requirements. The functional requirements are related to the knowledge or content of the ontology; therefore, they can be stated as competency questions (CQs) that the ontology should be able to answer. The CQs we defined for our ontology are related to the ML and explanation process components:

- CQ1. Which data were used to train the model?
- CQ2. How privacy-sensitive was the dataset?
- CQ3. How balanced are the data?
- CQ4. How were the data preprocessed?
- CQ5. What are the correlations of the input datasets?
- CQ6. What are the characteristics of the ML algorithm?
- CQ7. What is the logic behind the ML model?
- CQ8. Why did the model generate this output?
- CQ9. How was the ML model evaluated? What is the meaning of those metrics?
- CQ10. How were the explanations generated? How are the explanations presented to the user? How faithful are the explanations?
- CQ11. How general are the explanations? Do they apply to all instances?

The non-functional requirements are related to characteristics, qualities, and general aspects not related to the content [13]. We define the non-functional requirements of our ontology as follows:

- REQ1. Guarantee usability to data scientists and developers who want to understand the adequacy of the ML model and improve product efficiency, research, and new functionalities, helping understand the whole ML process and explanation process, possibly identifying where to make adaptations in the process;
- REQ2. Guarantee extensibility by defining a generic ML ontology that represents ML processes that tackle different problems besides classification and can be further adapted or specialized;
- REQ3. The ontology should be implemented using standard languages and tools;
- REQ4. Guarantee interoperability with already existing ontologies by grounding them into a foundational ontology.

In order to comply with REQ2 and considering the complexity of the ontology, we decided to structure the ontology into three modules: (1) a general module that represents general ML processes independently of the task or the learning type performed; (2) a specific module for supervised classification; and (3) an explanation module, which represents the post-hoc explanation process. Splitting the ontology into smaller parts allows the modeling problems to be tackled one at a time [13].

### 2.1.2. Knowledge Acquisition and Reuse

Knowledge Acquisition and Reuse are auxiliary processes defined in SABiO that assist ontology development [13]. Usually, Knowledge Acquisition occurs in the initial stages of ontology development to gather knowledge from different sources, while Reuse can be adopted in many opportunities to reuse already established conceptualizations.

This paper applies Reuse in the Knowledge Acquisition process by selecting an already existing domain ontology and a foundational ontology. First, the ML process and the post-hoc explanation process are defined with their components and what should be described, as represented in Figure 2.

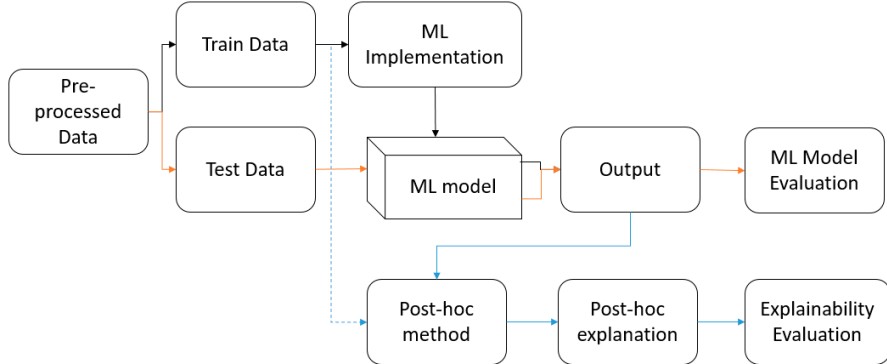

**Figure 2.** ML process with a training sub-process (black arrows), a testing sub-process (orange arrows), and a post-hoc explanation sub-process (blue arrows).

The Machine Learning Process: This process consists of preprocessing data, training, and testing the ML model. The preprocessed data determine the preprocessing steps, such as the cleaning process, feature extraction or dimensionality reduction methods applied with respective parameters, and the criteria to split training and testing data. Given the different nature and particularities of the available datasets, which may require diverse preprocessing steps to make it adequate for the ML, we assume that the data have already been preprocessed so that only the preprocessing steps taken are modeled in the ontology, without more details.

Training the ML model involves the training data, the ML implementation, the ML model, and the output. The training data represent the input data used to train the ML model. The ML implementation indicates the type and characteristics of the implemented algorithm, for example, Support Vector Machine (SVM) with the parameters used to train the model. The ML model (learned model) can be explained through the logic of the reasoning behind the decision-making process in general [7], which enables the user to understand the logic of the ML algorithm and the patterns observed by the ML model in the data. The description of the outputs obtained from the ML model is related to the explanation of a decision which refers to reasons that justify why a particular outcome was generated by the ML model.

Testing the ML model comprises the descriptions of the testing data and the ML model evaluation. The description of the evaluation is relevant because many metrics can be used to evaluate the ML models. Choosing the best metrics depends on the task that the ML model is expected to perform and its application. Evaluation descriptions provide information about the metrics and how the ML model performs with respect to these metrics.

The Post-hoc Explanation Process: This process comprises the explanation method, the explanation generated by the method, and the explainability evaluation. The method usually receives the output data from the ML model, and some of the methods also use the input training data to generate the explanation. The description of the post-hoc method indicates the method adopted and its characteristics; for instance, the scope of the method (local or global explanations), the format of the explanation it generates (tree, rules, decision table, images, text highlight, natural language, etc.), and whether the explanation is iterative or static. The explanation gives the logical reasoning behind the decision-making process, the patterns observed in the data, or provides means to justify the ML output. The evaluation contains information about the assessment of the explanations which can be evaluated in terms of effectiveness and user experience, such as the number of instances that the rules cover, and how faithful the explanations to the underlying black-box are.

### 2.1.3. Domain-Specific Ontology

An existing ontology was selected (ML-Schema [14]) as a starting point to develop our ontology to help the knowledge acquisition process, speed up the ontology development, and guarantee interoperability to existing applications that use the ontology [15].

ML-Schema (MLS) was chosen as the main reference to be reused and extended because it is already based on other well-known ontologies and contains many of the concepts necessary to represent the ML process. It is a well-known ontology for the ML domain which aims to stimulate the development of standards, achieve interoperability and reproducible research, cope and align with already existing ontologies, and support the needs of the ML area. MLS preserves the provenance of data and model, that is, metadata about their origin, derivation, or history. In ML workflows, it is useful to represent which data were used to train the ML model, where the data came from, and how they were preprocessed [14]. Therefore, MLS allows us to track the creation, editing, publication, and future reuse of data.

### 2.1.4. Foundational Ontology

Foundational ontologies define the basic concepts upon which any domain-specific ontology is built. By explicitly modeling the 'upper-level ontology', the top-level domain-independent ontological categories can be reused in domain-specific ontologies, guaranteeing semantic interoperability between them [16].

In this paper, we chose to adopt the Unified Foundational Ontology (UFO) [16] as suggested in SABiO [13]. UFO is based on two foundational ontologies, the GFO/GOL and OntoClean/DOLCE, offering a general foundational ontology to applications in conceptual modeling. UFO is divided into three sets: UFO-A, which is the UFO core, defines the things, sets, entities, individuals, and types; UFO-B defines the terms related to perdurants, such as events and states; and UFO-C defines terms related to beliefs, desires, intentions, social roles, and linguistic things, extending UFO-B with concepts such as action, activity, and communication [16,17]. UFO also defines two taxonomies, one with classes whose instances are individuals and another with classes whose instances are types, providing additional information about classes [18].

OntoUML is a language for ontology-driven conceptual modeling based on UFO [19]. It is built as an extension of UML (Unified Modeling Language), enabling conceptual models to be defined as fragments of UML class diagrams that are well-founded in UFO. This also facilitates the process of obtaining a well-founded operational ontology from the conceptual model by performing transformations. The resulting operational ontology is well-founded in gUFO, which is a lightweight UFO implementation that supports a subset of UFO-A and a minimal subset of UFO-B [18].

### 2.2. Ontology Development

Ontology development consists of capturing and formalizing the ontology with its modules and metadata. For this, the existing domain-specific ontology (MLS) has been first grounded in the foundational ontology (UFO), and the conceptual models of the three modules of the ontology were defined, namely the generic ML module, the specific ML module, and the explanation module. The design and implementation of the ontology have been carried out by a transformation from the final conceptual model to the operational ontology.

### 2.2.1. Grounding the MLS in UFO

In order to ground the MLS to UFO, first, a conceptual model of MLS using OntoUML [19] was developed. Using Visual Paradigm 16.3 and OntoUML plugin [20], we created a class diagram and assigned OntoUML class and relationship stereotypes to it, to perform a model transformation of the conceptual model into the operational version implemented in OWL (Ontology Web Language), which is a language to define ontologies, supported by gUFO.

However, grounding a domain ontology into a foundational ontology requires them to be aligned, which leads to concerns such as how to overcome differences in expressiveness that can exist between the ontologies and how to accommodate for the different philosophies behind them [21]. Therefore, to identify the best stereotypes to assign to the components of the diagrams, first the OntoUML stereotypes were studied and analyzed together with their matching in the gUFO structure to define a mapping between them. The MLS qualities, processes, and information entities have been grounded by creating classes in the class diagram and assigning the OntoUML class stereotypes to them according to the chosen components of the gUFO taxonomy of types.

The MLS relations have been analyzed and considering the types they connect, adequate relationships in OntoUML have been selected, which were then transformed into adequate object properties in gUFO, as shown in Table 1. For example, the process Run of MLS is defined as an Event in OntoUML and gUFO taxonomies of types and individuals because it consists of a process that happens in time. An Experiment is a collection of runs with no change in membership since specific runs are part of an experiment, therefore, an Experiment is grounded also as an Event. Similarly, a Study is a collection of Experiments, defined also as an Event.

**Table 1.** Examples of the correspondences of ML-SCHEMA components to OntoUML and gUFO taxonomy of individuals and taxonomy of types.

| ML-SCHEMA Element | OntoUML Stereotype | gUFO Individual | gUFO Type |
|---|---|---|---|
| Processes (Examples: Study, Experiment, Run)Study | Event | Class Event | Event Type |
| Qualities (Examples: DataCharacteristic ImplementationCharacteristic, ModelCharacteristic) | Quality | Class Quality | Kind |

MLS qualities such as DataCharacteristic, ImplementationCharacteristic, and ModelCharacteristic are intrinsic aspects that are measurable or may be used to compare individuals, for example, the number of features a dataset has, or a characteristic of an implementation that differentiates it from the others, being defined as Quality in OntoUML and in gUFO's taxonomy of individuals, which corresponds to Kind in gUFO's taxonomy of types.

### 2.2.2. General Machine Learning Module

The general ML module of our ontology has been developed by aligning the MLS concepts grounded in gUFO with the ML process. MLS focuses on the data generated in ML workflows, reflecting the OpenML structure, but does not represent the nature of ML processes, resulting in semantic gaps. For example, we missed some concepts to organize the Runs into Experiments, arranging these Runs in a sequence to be executed. Thus, the event Workflow Execution (WFExecution) has been introduced, which represents processes that belong to an Experiment, and is composed of a series of Operations and executes Workflows, which organize sequentially the implementations that are executed by these operations.

In addition, the lack of cardinality in MLS can generate misconceptions, for instance, indicating that a Run generates both an evaluation and a model. However, a dedicated Run usually outputs an ML model, and another Run dedicated to evaluating the model outputs the evaluation of the ML model in terms of the metrics. Other types of Runs generate other outcomes, such as preprocessed data or predictions. Hence, we included the Output concept to generalize the output generated by the operations, which can then be specialized into ML model, evaluation, etc.

After properly aligning MLS with the ML process and defining the concepts and relations of the generic ML module, the axioms and constraints necessary for this ontology module have been defined, and after that, the conceptual model was developed. We defined restrictions concerning the cardinality of the properties between the concepts, disjoint classes, and the sequence that each operation needs to be executed, formalizing them in OWL2, which is the most recent OWL version.

### 2.2.3. Specific Machine Learning Module

The general ML module can be further specialized by considering the different operations that are performed in the ML classification process, taking into account their different participants. The tasks involved in the classification process usually consist of preprocessing the data, training, testing, and evaluating the ML model. Each task is represented as a subclass of the operation class in the specific ML module, with the participants and the artifacts that are involved in the operation.

### 2.2.4. Explanation Module

The ontology containing the generic and specific modules have been extended by adding the explanation module, which represents the post-hoc explanation process by adding the Explain and Evaluate Explanation operations to their corresponding participants.

### 2.2.5. Metadata

The ontology has been complemented with semantic information that comprises qualities, reified quality values, and annotations in the ontology. In gUFO, qualities are intrinsic aspects that are measurable by receiving a literal value, while reified quality values are abstract individuals that can use pre-defined data to provide the value of the quality, instead of literals. The reified quality values have been instantiated in the gUFO taxonomy of types as abstract individual types and the qualities as kinds and subkinds.

### 2.2.6. Ontology Design and Implementation

The ontology design and implementation steps aim at generating an operational version of the ontology. In the design step, the technical aspects of the ontology and the implementation environment were defined. Since the OntoUML plugin is based on UFO and it already supports the transformation of the conceptual model to the implementation language, the gap between the conceptual models and the operational is shortened, so that this step could be automated.

The implementation step consists of implementing the ontology in the operational language. This was performed by executing a transformation of the OntoUML conceptual model and exporting it to OWL in Turtle (Terse RDF Triple Language) format, which represents the ontology in the RDF data model. We then opened the exported file using Protégé and made further manual adjustments.

## 3. Case Study

The COVID-19 scenario was adopted as a case study to illustrate and validate our ontology. Supervised learning algorithms that perform classification operations were used to predict mortality among infected patients and existing explanation methods were applied to generate explanations that intend to make the steps and the logic of the algorithm clear to data scientists to improve the model. Based on the results, instances have been created to populate the ontology, validating it and refining it if necessary. The ML algorithms and supporting procedures were implemented in Python scripts.

### 3.1. Data Description

We used an epidemiology dataset of people tested for COVID-19 in Mexico [22]. Health care data are sensitive by definition because they contain personally identifiable information, increasing the need of a trustworthy machine learning model. The dataset has

566.602 instances with 23 features, containing demographic data such as age and gender of the patient, pre-existing conditions, for instance, diabetes, chronic obstructive pulmonary disease (COPD), asthma, immunosuppression, hypertension, obesity, pregnancy, chronic renal failure, other prior diseases, and whether the patient used tobacco. It indicates if the patient was hospitalized, had pneumonia, needed a ventilator, was treated in an intensive care unit (ICU), but also the result of the Reverse Transcription Polymerase Chain Reaction (RT-PCR) test, and the date when the patient deceased, if applicable.

### 3.2. Experiments

Two ML experiments were defined based on the available data. The goal of the first experiment was to classify the mortality of the confirmed cases and to understand the ML model using the RIPPER (Repeated Incremental Pruning to Produce Error Reduction) [23]. According to Martens et al. [24], RIPPER can be used to extract human-comprehensible descriptions from opaque models, based on decision rules in the format of IF-THEN statements, which are considered one of the most interpretable statements since this structure semantically resembles natural language [25]. Therefore, the opaqueness of inscrutable ML models can be remedied by extracting rules that mimic the black-box as closely as possible, since some insight is gained into the logical workings of the ML model by obtaining a set of rules that mimic the model's predictions [24].

The second experiment was similar to the first, but it used Local Interpretable Model-agnostic Explanations (LIME) [3] to generate explanations. LIME is one of the most popular solutions in the academic community to ML explainability [26]. It consists of a post-hoc model-agnostic tool that identifies an interpretable model that is locally faithful to the black-box classifier.

Each experiment involved two main workflows: the ML Workflow and the Explanation Workflow. The second experiment reused the ML Workflow of the first one.

In order to describe the experiment using the ontology, we included instances describing the experiments and their characteristics, the goal of the experiment, the workflows involved, and metrics to evaluate the ML models. The goal of classifying mortality is a binary classification, requiring the ML model to be evaluated using metrics such as accuracy, sensitivity, and specificity. The link to the conceptual models, ontology, and the code in Python to the experiments is available in Supplementary Materials.

### 3.3. The Machine Learning Workflow

The ML Workflow consists of the steps of preprocessing the input dataset, training, and testing the ML model.

#### 3.3.1. Data Preprocessing

The first step of the ML Workflow consists of preprocessing the dataset. From the dataset available we used a small sample of patients that entered the hospital between the first two days of June 2020 and that were diagnosed with positive results for COVID-19. Patients with unknown information were manually removed, keeping only the ones that indicated the presence or absence of conditions. The date of death was converted to a binary column indicating one if the patient was deceased and zero if she/he recovered.

After making the dataset fit for processing in the Python environment, we removed the columns id (patient's identifier), the date that the patient started feeling symptoms, if the patient had contact with other COVID-19 cases, the result of the RT-PCR test (since they were already manually filtered), and pregnancy. The column 'inmsupr' was renamed to 'immunosuppression'. The final dataset remained with 9.451 cases and was split into 70% for training and 30% for testing. No feature extraction or dimensionality reduction methods were adopted.

The preprocessing steps performed in Python and the parameters used to the preprocessing functions are included in the ontology by adding instances to represent the scripts executed by the preprocessing operation. Manual steps are disregarded. Differences

between the original dataset and the preprocessed dataset can be identified, such as the difference between the number of instances of each dataset and, after deleting some columns from the original data, the presence of some features as components of the input dataset and their absence in the preprocessed datasets.

### 3.3.2. Data Analysis

A dataset analysis has been performed to obtain a more detailed description of the datasets. For this, we chose to adopt the explainability tool EthicalML-XAI [27] that helps identify imbalances between features across classes and provides functions to identify correlations between features in the dataset. This tool enables correlations between variables to be detected and unexpected correlations to be identified (for example, intubation should be related to ICU), allowing the user to identify possible problems in the dataset.

The imbalances and correlations are included in the ontology as characteristics of features that belong to the training dataset.

### 3.3.3. Machine Learning Model Training

The preprocessed training data were used to train a black-box model to classify the mortality of confirmed COVID-19 cases. We used SVM as the black-box algorithm, given its popularity in making classifications due to its ability to capture non-linearity [24], and its common application in COVID-19 detection in the literature, as found in [28–32].

The ML training operation is described in the ontology as the operation that receives the training dataset as input and generates the SVM model as output. It executed the SVM implementation of the SVM algorithm provided by the scikit-learn library in Python [33], using a linear kernel type as a parameter. The ML algorithm is described in terms of the author of the algorithm; the algorithm transparency, defined by Molnar [25] as the description of how the algorithm usually works; the type of data adequate for this algorithm; the involved ML technique, in this case, classification; and the learning type, which in this example is supervised learning.

### 3.3.4. Machine Learning Model Evaluation

After the ML model was trained, a step was performed to evaluate it according to the metrics established by the goal. For this, the test set was classified using the Test (Predict) operation that executes the Predict Implementation. This implementation called the SVM model function to generate the predictions. These predictions together with the original labels of the test set were then used as input for the Evaluate Model operation.

The Evaluate Model operation executed implementation of the classification report provided by the scikit-learn library in Python [33], which already contains usual metrics for classification including accuracy, specificity, and sensitivity. SVM achieved an accuracy of 0.91 in the test set, specificity of 0.92, and sensitivity of 0.64. The metric values and the description of each metric are included in the ontology.

### 3.4. The Explanation Workflow

The explanation workflow represents the explanation process, which consists of two main steps, namely the generation of explanations and the evaluation of these explanations. The objective of the first step is to generate explanations from the black-box model using post-hoc methods that identify the behavior of the ML model or try to explain results. The second step is to evaluate the generated explanations, which is necessary since post-hoc solutions generate hypotheses to explain the black-box model.

In our case study, the explanation workflow uses the method RIPPER [23] in the first experiment and LIME in the second experiment to generate explanations.

### 3.4.1. Rule Extraction with RIPPER

The opaqueness of SVM models can be remedied by extracting rules that mimic the black-box as closely as possible since some insight is gained into the logical workings of

the SVM by obtaining a set of rules that mimic the model's predictions [24]. Therefore, rule extraction can be performed to understand the classifications of the SVM, opening up the black-box.

In order to extract rules from the SVM, RIPPER [23] was applied to the dataset with labels predicted by the SVM. This algorithm extracts a rule set for the classification of the COVID-19 mortality class, as depicted in Figure 3. The rule set is composed of disjunctive rules that lead to the classification of the positive class, in this case, mortality. Each line is a rule consisting of a conjunction of clauses. We interpreted the rule set as follows: if the first rule applies to the instance, for example, the patient has an age between 66 and 99 years (age = 66–99), has pneumonia (pneumonia = 1), has hypertension (hypertension = 1), is hospitalized (hospitalized = 2), and is a man (sex = 2), then he will be classified with a high chance of mortality. If the patient did not fit in this first rule, we tried the second, and so on, until the last rule. If none of the rules applied to the patient, he/she has been classified with the negative class, that is, recovery.

```
FINAL RULESET:
[[age=66-99 ^ pneumonia=1 ^ hypertension=1 ^ hospitalized=2 ^ sex=2] V
[pneumonia=1 ^ age=66-99 ^ icu=2 ^ intubed=1] V
[pneumonia=1 ^ age=66-99 ^ icu=2 ^ hypertension=1 ^ asthma=2 ^ copd=1] V
[pneumonia=1 ^ age=66-99 ^ icu=2 ^ hypertension=1] V
[pneumonia=1 ^ intubed=1 ^ icu=2] V
[pneumonia=1 ^ age=66-99 ^ icu=2 ^ sex=2] V
[renal_chronic=1 ^ pneumonia=1 ^ hospitalized=2 ^ sex=2] V
[pneumonia=1 ^ age=66-99 ^ intubed=1] V
[pneumonia=1 ^ age=66-99 ^ icu=2 ^ cardiovascular=1] V
[pneumonia=1 ^ age=66-99 ^ renal_chronic=1] V
[hospitalized=2 ^ renal_chronic=1 ^ other_disease=1] V
[pneumonia=1 ^ renal_chronic=1 ^ age=51-57] V
[icu=2 ^ age=66-99 ^ renal_chronic=1] V
[pneumonia=1 ^ hypertension=1 ^ age=57-66 ^ sex=2 ^ intubed=1] V
[age=57-66 ^ hypertension=1 ^ pneumonia=1 ^ diabetes=2 ^ sex=2 ^ tobacco=2] V
[intubed=1 ^ icu=2]]
```

**Figure 3.** Rule set extracted using RIPPER to classify mortality in COVID-19 cases.

In the ontology, RIPPER is described in terms of its author, the source reference paper, and further information to indicate that it consists of a global model-agnostic method that explains the ML model and is unfaithful to the underlying model because it extracts rules from the training examples, not directly from the ML model. The explanations generated by the algorithm are also described in terms of its format, i.e., the method generates static explanations in the format of rules, and the explanation explains the logic behind the whole ML model instead of explaining only one instance. Its implementation in Python was provided by the Wittgenstein library [34].

### 3.4.2. LIME Explanations

The post-hoc model-agnostic tool LIME [3] was also applied to identify the impact of each input variable on the classification. First, the Submodular Pick Module (SP-LIME) was used to generate explanations that show the positive and negative impact of the input variables for each class. SP-LIME selects a set of representative instances, that is, non-redundant and globally representative, and their explanations, enabling us to verify if the model behaves adequately as expected.

The same explainer used in SP-LIME was applied to generate explanations of specific instances. Figure 4 depicts an explanation obtained with LIME for a single patient, showing the high probability of recovery predicted by the trained SVM and the impact of each variable on each class, with intubation having the highest impact on the classification.

In the ontology, LIME is described in terms of its authors, the source reference paper, and further information to indicate that it is a local model-agnostic method that explains the ML model and the outputs, and that it is locally faithful to the underlying model. The explanations generated by the algorithm are also described regarding its format, i.e., the method generates static explanations in the format of the weight impact of the variables. Its implementation in Python is provided by the LIME library [3].

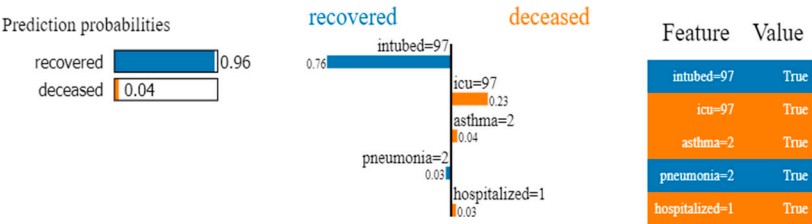

**Figure 4.** Explanation generated by LIME for one instance, indicating a higher probability of recovery and the weights of the most impacting features for each class.

### 3.4.3. Explanation Evaluation

The generated explanations were evaluated in the Evaluate Explanation operation. The RIPPER rules can be evaluated considering their coverage and the accuracy achieved by applying the rule set to predict the classes of a data set. Using the test set, an accuracy of 0.9 was achieved, meaning that 90% of the test examples had their classes correctly predicted using the rules obtained by the RIPPER method.

The SP-LIME explanations can also be evaluated considering the coverage of the explanations. Using the training set, we obtained the number of instances covered by the explanation. These characteristics are included in the ontology to describe the components of the explanation evaluation.

### *3.5. Ontology Evaluation*

The last step of ontology development is ontology evaluation. The SABiO evaluation process prescribes ontology verification, which aims to ensure that the ontology complies with the specifications previously defined, such as the ontology requirements, and ontology validation, which aims to ensure that the ontology fulfills its intended purpose [13].

To verify the ontology, we first analyzed if the ontology met its requirements. Ontology validation was conducted by creating instances of the ontology and querying the implementation environment according to the CQs, checking if the obtained results were the expected outputs [13].

### 4. Results

The conceptual model of the ontology was developed in an OntoUML diagram that expresses typed relations between components, cardinality constraints for the relations, and constraints related to which element can be connected to others, formalizing specifications and axioms. The diagram is shown in Appendix A and contains three modules, namely (1) the General ML Module, represented within the grey UML package; (2) the Specific ML Module, in the yellow UML package; and (3) the Explanation Module, in the green UML package.

The General ML Module is generic and has been developed to be reused to represent other ML processes and further adapted or specialized. Study has a purpose and consists of experiments, which have Workflow Executions. Workflows are composed by operations. Experiment has a goal, which is addressed by the ML Algorithm, e.g., Support Vector Machine (SVM) and Artificial Neural Network (NN). The operation is an event that receives data input, executes an implementation of the algorithm, receives parameter settings, and creates outputs, which can be an ML Model or a Model Evaluation.

The Specific ML Module considers the different operations that are performed in the ML classification process, which consists of preprocessing the data, training the ML model, testing the ML model, and finally evaluating the ML model. Each of them is represented as a subclass of operation with the participants and the artifacts that are involved in the operation. First, Preprocess takes place to make the input data suitable to train and test the ML model, receiving as input InputData, and executing a PreprocessImplementation, which in turn implements a PreprocessingAlgorithm. The output of this operation is PreprocessedData, which can be specialized into the subclasses TrainData and TestData.



TrainData participates in a Train operation that fits the ML model, executing an ML implementation and generating a fitted MLModel. The Test (Predict) receives TestData and executes a PredictImplementation, calling a fitted MLModel to predict output results. The prediction for each instance is represented by ResultInstance. Finally, an EvaluationModel operation for labeled data receives input results and compares them with TestData. It executes an EvaluateModelImplementation that implements an EvaluationProcedure, such as cross-validation or leave-one-out, taking into account EvaluationMeasures that need to be evaluated. This operation generates a ModelEvaluation that contains the values for the measurements specified by an EvaluationMeasure.

The Explanation Module represents the post-hoc explanation process by adding the Explain and EvaluateExplanation operations with their corresponding participants. The first operation, Explain, aims to generate explanations by using results generated by a Test (Predict) operation and in some cases also using PreprocessedData. It executes the implementation of an ExplainableAlgorithm to generate an explanation, which can be classified as MLExplanation if it aims to explain the ML model, or as ResultExplanation, if it explains only a resulting instance. The relator ExplainsModel, between an MLModel and an Explanation, allows the logic behind an ML model to be obtained after the post-hoc method is applied. The operation EvaluateExplanation is related to the assessment of the explanations, executing an EvaluateExplanationImplementation to generate an ExplanationEvaluation.

The operational ontology was obtained using the OntoUML plugin to transform the conceptual model to OWL, which can be manipulated in Protégé to populate the ontology and to make further manual adjustments, such as including named relationships. This is necessary because the transformation only automatically transforms native relationships of OntoUML to their correspondents in gUFO.

By evaluating the ontology, we assessed if the ontology meets its requirements. For the ontology quality attributes, our ontology proposes a more generic module that can be further adapted and specialized, satisfying REQ2. For the project requirements, our operational ontology is implemented in Protégé represented in OWL, satisfying REQ3. Considering the intended user-related requirements, our ontology is grounded in gUFO, fulfilling REQ4. With the aid of Protégé Reasoner and gUFO Protégé Plugin [35], we also checked the quality and correctness of the ontology implementation to assess if it meets the language specifications in terms of not having inconsistencies and satisfying the rules of a gUFO-based ontology.

REQ1 states that the ontology is adequate for data scientists and developers to understand the adequacy of the ML model and make adaptations and improvements, and is related to ontology validation. In order to ensure that the ontology fulfills its intended purpose, ontology validation has been conducted by creating instances of the ontology and querying the implementation environment according to the CQs, checking if the obtained results are the expected outputs [13]. For this, the instances have been created based on the case study and the CQs were implemented as queries using SPARQL, which is a query language applicable to RDF data models. RDF is the underlying technique used in OWL/OWL2.

For example, for CQ1 ("Which data were used to train the model?"), Figure 5 shows the SPARQL code used to query the ontology to obtain information about the data used to train the ML model in the first experiment. The same code can be also applied to query the ontology for the second experiment by substituting the value "Experiment1" to "Experiment2".

The queries retrieve information about the data period, the source, when the source was accessed, the number of instances and features, and a description of the data, as expected. By performing different queries for each CQ, we demonstrated that the ontology is able to answer all CQs, which is the first step to validate the ontology.

```
PREFIX rdf: <http://www.w3.org/1999/02/22-rdf-syntax-ns#>
PREFIX owl: <http://www.w3.org/2002/07/owl#>
PREFIX rdfs: <http://www.w3.org/2000/01/rdf-schema#>
PREFIX xsd: <http://www.w3.org/2001/XMLSchema#>
PREFIX gufo: <http://purl.org/nemo/gufo#>
PREFIX XMLo: <https://example.com#>
SELECT ?experiment ?WFExecution ?preprocessOperation ?inputdata ?characteristic ?property ?values
WHERE {
    {
        ?characteristic  ?property ?values.
        ?characteristic gufo:inheresIn ?inputdata.
        ?inputdata rdf:type XMLo:InputData.
        ?inputdata gufo:participatedIn ?preprocessOperation.
        ?preprocessOperation rdf:type XMLo:Preprocess.
        ?preprocessOperation  gufo:isEventProperPartOf  ?WFExecution.
        ?WFExecution gufo:isEventProperPartOf ?experiment.
        FILTER(?property not in (gufo:participatedIn, rdf:type, gufo:inheresIn)).
    }
    UNION{
        ?inputdata ?characteristic ?values.
        ?inputdata rdf:type XMLo:InputData.
        ?inputdata gufo:participatedIn ?preprocessOperation.
        ?preprocessOperation rdf:type XMLo:Preprocess.
        ?preprocessOperation  gufo:isEventProperPartOf  ?WFExecution.
        ?WFExecution gufo:isEventProperPartOf ?experiment.
        FILTER(?characteristic not in (gufo:participatedIn, rdf:type)).
    }
    FILTER(?experiment=XMLo:Experiment1).
    FILTER(?values != owl:NamedIndividual)
}
```

**Figure 5.** SPARQL query for CQ1 for the first experiment.

## 5. Conclusions

In this paper, we propose an ontology to describe the main components of the ML process and post-hoc explanation process, providing means that enable a user to have a holistic understanding of why the ML model arrived at these specific results. In the literature, different ontologies for data mining and ML can be found, but currently, to our best knowledge, there is no ontology to represent ML and explanation processes that also aim to achieve explainability by focusing on a complete overview of all components of these processes. Our ontology was developed following the best practices by adhering to the SABiO methodology and explores state-of-the-art technologies for ontology engineering, for instance, the OntoUML plugin and the UFO plugin for Protégé. The ontology is based on existing ontologies, but it also considers different points of view from domain experts. We aligned MLS, which is the main domain-specific ontology that inspired the development of our ontology, with the ML process and grounded it in a foundational ontology, making the ontology interoperable with existing ontologies that follow UFO.

The process of grounding the domain ontology presented some challenges because it required the alignment of ontologies that follow different philosophies and deep knowledge of their concepts. OntoUML has been fundamental to facilitating the grounding process. However, it required the use of gUFO, lacking the support of UFO-C, which has elements that could be used in our ontology to represent, for example, actions and goals. After grounding MLS in gUFO, our ontology was structured using modules, and descriptors of each element considered necessary to describe the processes were added to the ontology.

The ontology was then evaluated by using the OntoUML plugin for Visual Paradigm, which was used to build the conceptual model and also provided verification, showing that our ontology is sound. After the transformation of the conceptual model to the gUFO-based operational ontology, we used Protégé to create individuals and validate the ontology. The reasoner and the UFO plugin available to Protégé also assisted the evaluation process by verifying consistency and ensuring that the rules of our gUFO-based ontology were satisfied.

A case study was defined by considering the scenario of the COVID-19 pandemic, training an SVM model with data of infected patients to predict the mortality, and applying existing explanation methods to get feature correlations among the training data. We also applied post-hoc explanation methods to generate explanations concerning the behavior of the ML model, generating rules with RIPPER and obtaining the impact of each variable on the result with LIME. With the application of these methods, we obtained different explanations of the data and the trained ML model, which helped us better understand the logic behind it. The ontology was then populated with instances that describe the case study, which helped identify necessary changes. This also enabled the ontology to be queried, retrieving information for each CQ that ensured that the ontology fulfilled its intended purpose, leveraging the post-hoc explainability.

Our ontology has been designed to enable the description of different kinds of ML experiments, with different ML algorithms, and it is modularized to enable the general ML

module to be extended and used for other purposes. It also aims to enable the description of different post-hoc explanation methods. The information that describes ML algorithms, explanation methods, metrics, etc., is expected to be easily reused.

Furthermore, making the logic behind the ML model and the whole ML and explanation process clearer can help to ensure better understandability and trust. This is needed in many situations, such as the case study explored by this paper, which worked with private and sensitive patients' data and used the ML to predict the severity of a disease.

Concerning future work, the process of feeding the ontology with instances can be automated by adopting technologies that can create individuals in the ontology while conducting the experiments, which should facilitate and encourage the use of the ontology. The ontology can be further tested with other ML models, other types of data, and different post-hoc explanation methods, verifying if they have peculiarities that should be modeled or extending the vocabulary, and verifying that the ontology, especially the Generic ML Module, can be reused and extended for other purposes. Moreover, we can involve data scientists and developers to validate the use of the ontology as a tool that complements the explanations and helps understand the adequacy of the ML Model.

The ontology can also be extended to represent manual preprocessing steps or preprocessing steps that are machine learning models themselves, for example, when applying ML to execute dimensionality reduction or feature extraction. Furthermore, the dataset correlations could be tracked during the whole ML process, and the explanation can be described as mappings between the inputs and the output. Aspects related to data sensitivity and privacy can also be further explored.

Considering the novelty of exploiting the post-hoc explanation methods and representing them using ontologies, ante-hoc approaches could also be evaluated in terms of whether they can be generically represented by a single ontology, taking into account the diverse approaches that require changes in the implementation of ML algorithms. If this is the case, the ontology presented in this work can be analyzed regarding its reusability and extendibility to cover also ante-hoc approaches, or if it is necessary to develop a new ontology.

Finally, we intend to include requirements related to data federation, considering that an ML model can be trained across multiple decentralized devices without exchanging data samples, as in federated (collaborative) learning, which may lead to trust issues.

**Supplementary Materials:** The conceptual models and ontology are available online at https://github.com/pNakagawa/ExplainableMLOntology, accessed on 30 September 2021.

**Author Contributions:** Conceptualization, P.I.N., J.L.R.M., L.F.P. and L.O.B.d.S.S.; methodology, P.I.N., J.L.R.M., L.F.P. and L.O.B.d.S.S.; software, P.I.N.; validation, P.I.N., J.L.R.M., L.F.P. and L.O.B.d.S.S.; formal analysis, P.I.N., J.L.R.M., L.F.P. and L.O.B.d.S.S.; investigation, P.I.N.; resources, P.I.N.; data curation, P.I.N.; writing—original draft preparation, P.I.N.; writing—review and editing, L.F.P., J.L.R.M., and F.B.; supervision, L.F.P., J.L.R.M., L.O.B.d.S.S. and F.B. All authors have read and agreed to the published version of the manuscript.

**Funding:** The author (P. I. Nakagawa) was supported by the Orange Tulip Scholarship to fund its master studies.

**Institutional Review Board Statement:** Not applicable.

**Informed Consent Statement:** Not applicable.

**Data Availability Statement:** Data obtained from M. R. Franklin, "Kaggle: Mexico COVID-19 clinical data," 6 May 2020 (https://www.kaggle.com/marianarfranklin/mexico-COVID19-clinical-data/metadata, accessed on 30 September 2021). This data is processed from Secretaría de Salud, Datos Abiertos Dirección General de Epidemiología (https://www.gob.mx/salud/documentos/datos-abiertos-152127, accessed on 30 September 2021).

**Conflicts of Interest:** The authors declare no conflict of interest.

## Appendix A

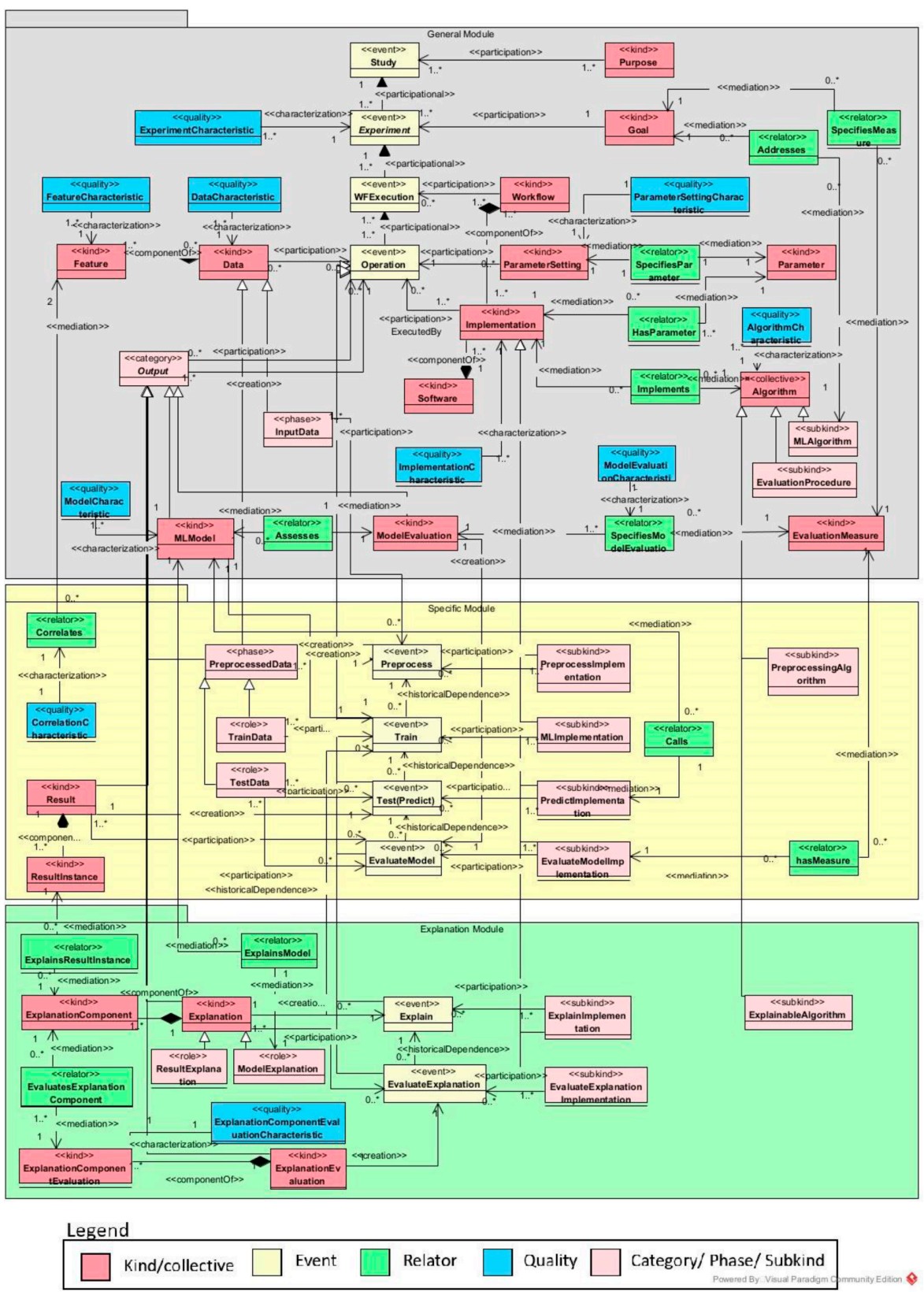

**Figure A1.** Conceptual model of the ML explanation ontology that is composed of the general ML module (grey region), the specific classification module (yellow region), and the explanation module that represents the post-hoc explanation process (green region).

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
