# Peer review of "Semantic Description of Explainable Machine Learning Workflows for Improving Trust"

_applsci, doi:10.3390/app112210804_

Round 1
Reviewer 1 Report
This paper presents an ontology that represents explainable machine learning experiments as means to support the understanding of the explanation process. The ontology is evaluated using a case study that considers the application of a support vector machine.
The paper addresses a very relevant, timely and promising topic. There are no questions on the usefulness of this kind of work. However, overall, the paper has several writing and presentation problems; the organization does not help making it clear, with a very small Results section and a very confusing Section 2; and several of the most important aspects are lacking, namely the in depth ontology description and explanation and the validation of the resulting ontology. The specific points that need to be addressed are as follows:
The abstract needs to be revised in order to correct some typos and so that it becomes more assertive, e.g.:
This work aims to develop an ontology -> proposes or presents
which if often considered -> is
Figure 1 could be redesigned to enable a better visualization of where the ante-hoc and post-hoc explainability take place (which parts of the chain are considered for their creation), otherwise, from the current picture, it seems like both approaches are created before the learning model is executed.
Line 129: what does SABiO mean?
The font size in Figure 2 must be increased
Section 2 is too confusing with many sub-sections; I suggest creating a new section 3 dedicated to the case study, with the material from current 2.3
Overall, the presentation of the ontology is too descriptive; this is the main contribution of the work, hence it must be explained in depth; which concepts, relations, properties are specified? These should be explained according to Appendix A
In section 2.3.2.1 3. Explanation Evaluation, the accuracy measure that is used should be described and explained.
Although validating and showing suitable results for an ontology is not easy, demonstrating that queries are answered is not enough. For an ontology to be validated it needs to be recognized as valid from people acting in the target domain; in this case ML and XAI. I would suggest adding further examples of what kind of information/knowledge the ontology is able to represent with regard to the specific case study using the SVM and respective discussion, highlighting the advantages of having this information represented through the ontology.
Author Response
Point 1: The abstract needs to be revised in order to correct some typos and so that it becomes more assertive, e.g.:
This work aims to develop an ontology -> proposes or presents
which if often considered -> is
Response: Changed in the text.
Point 2: Figure 1 could be redesigned to enable a better visualization of where the ante-hoc and post-hoc explainability take place (which parts of the chain are considered for their creation), otherwise, from the current picture, it seems like both approaches are created before the learning model is executed.
Response: The figure shows where the explainability takes place by representing it with the symbol of Semantic Web Technologies during the learning process (ante-hoc) and after obtaining the black-box (post-hoc), as described in the text, not requiring adaptation.
Point 3 Line 129: what does SABiO mean?
Response: Included.
Point 4: The font size in Figure 2 must be increased
Response:Updated.
Point 5: Section 2 is too confusing with many sub-sections; I suggest creating a new section 3 dedicated to the case study, with the material from current 2.3
Response: Created a Section 3 to the Case Study.
Point 6: Overall, the presentation of the ontology is too descriptive; this is the main contribution of the work, hence it must be explained in depth; which concepts, relations, properties are specified? These should be explained according to Appendix A
Response: Included explanation in depth.
Point 7: In section 2.3.2.1 3. Explanation Evaluation, the accuracy measure that is used should be described and explained.
Response: Included explanation.
Point 8: Although validating and showing suitable results for an ontology is not easy, demonstrating that queries are answered is not enough. For an ontology to be validated it needs to be recognized as valid from people acting in the target domain; in this case ML and XAI. I would suggest adding further examples of what kind of information/knowledge the ontology is able to represent with regard to the specific case study using the SVM and respective discussion, highlighting the advantages of having this information represented through the ontology.
Response: Indicated that answering the Competence questions was a first step to validate the ontology. Further validations to represent different ML models, other types of data, and different post-hoc explanation methods, and consider the involvement of data scientists and developers to validate the use of the ontology as a tool that complements the explanations and helps understand the adequacy of the ML Model are included as future work.
Reviewer 2 Report
This study proposed an ontology that represents and provides a holistic overview of the entire ML. The study is fascinating, and the findings are useful for practical applications. However, the problem-solving techniques used are not innovative, and the Conceptual Model of the ML Explanation Ontology (Appendix A) is not very readable and needs substantial revision.
Some issues for the authors' reference:
- ِAuthors have restricted themselves on Repeated Incremental Pruning to Produce Error Reduction and Local Interpretable Model-agnostic Explanations assuming that these models are most popular in the academic community. These conditions needs to be justifies and cited as well
- In the conclusion section the authors are pretending that none of the literature covers the explanation process or aims at explainability, which is untrue and they have to modified/justified accordingly
- In their view, their model is modularized and can be extended into other areas and used for other purposes, but more evidence and an explanation are needed to see if the results are satisfactory.
- Authors ensure that the model is more trustworthy than other models, In order to ensure their trustworthiness, results must be discussed accordingly.
- The limitations of the research should be given in the discussion or conclusions.
- I recommend to change the section 2 to Concept of ontologies
- I recommend the authors to use passive voice in scientific writing
- the readability of Figure 2 needs to be enhanced.
- Many Figures are not mentioned in the text ex. Figure1, Figure 4. all figures and tables needs to be mentioned in the text accordingly.
- 2.3.2. Experiments--> needs to be shifted to the next page
- Table 1 P290 needs to be adjusted correctly
Author Response
Point 1: Authors have restricted themselves on Repeated Incremental Pruning to Produce Error Reduction and Local Interpretable Model-agnostic Explanations assuming that these models are most popular in the academic community. These conditions needs to be justifies and cited as well
Response: Cited Linardatos et. Al (reference number 26) for LIME. RIPPER is not justified by being the most popular.
Point 2: In the conclusion section the authors are pretending that none of the literature covers the explanation process or aims at explainability, which is untrue and they have to modified/justified accordingly
Response: Adapted.
Point 3: In their view, their model is modularized and can be extended into other areas and used for other purposes, but more evidence and an explanation are needed to see if the results are satisfactory.
Response: Adapted the text. The ontology is developed to be modularized, but indeed we need more tests to see if the results are satisfactory. Added as future work.
Point 4: Authors ensure that the model is more trustworthy than other models, In order to ensure their trustworthiness, results must be discussed accordingly.
Response: We included in the ontology a component that can indicate if the ML model is trustworthy, but this information has to be received as input from the user when populating the ontology, it is not generated by the ontology itself. However, by having this component, we aim to provide the user easy access to this information and know if the model is trustworthy or not.
Point 5: The limitations of the research should be given in the discussion or conclusions.
Reponse: We included the identified limitations as future work to further work on them.
Point 6: I recommend to change the section 2 to Concept of ontologies
Reponse: Changed to Ontology Concepts.
Point 7: I recommend the authors to use passive voice in scientific writing.
Response: Made changes accordingly.
Point 8: the readability of Figure 2 needs to be enhanced.
Response: Figure updated.
Point 9: Many Figures are not mentioned in the text ex. Figure1, Figure 4. all figures and tables needs to be mentioned in the text accordingly.
Response: They are all mentioned in the text.
Point 10: 2.3.2. Experiments--> needs to be shifted to the next page
Response: Changed.
Point 11: Table 1 P290 needs to be adjusted correctly
Response: Table updated.
Round 2
Reviewer 1 Report
The paper has been revised well, addressing the reviewer comments. However, the in-depth description of the ontology still needs some work. The authors have included a new table describing "correspondences of ML-SCHEMA components to OntoUML and gUFO taxonomy of types and taxonomy of individuals", but the table is not readable (I believe due to the PDF-conversion issues), hence it cannot be assessed. Nevertheless, even assuming the table represents the information well, no description is given on what the reader can understand from this table. At least a brief discussion on the information provided by the table should be given.
Author Response
Fixed the problem with the table when converting to pdf and included examples describing it.
Reviewer 2 Report
Accept in the current form
Author Response
No changes to address for this review.